High moon brightness and low ambient temperatures affect sloth predation by harpy eagles

de Miranda Everton B.P. 1 mirandaebp@gmail.com
Kenup Caio F. 2
http://orcid.org/0000-0001-5896-6105 Campbell-Thompson Edwin 3 4
Vargas Felix H. 4
Muela Angel 4
Watson Richard 4
http://orcid.org/0000-0002-1588-8765 Peres Carlos A. 5 6
http://orcid.org/0000-0001-8334-1510 Downs Colleen T. 1
1 School of Life Sciences, Centre for Functional Biodiversity, University of KwaZulu-Natal, Pietermaritzburg , Pietermaritzburg, KwaZulu-Natal , South Africa
2 Wildlife and Ecology Group, Massey University , Palmerston North , New Zealand
3 Fundación Aguilas de Los Andes , Pereira , Colombia
4 The Peregrine Fund , Boise, ID , USA
5 School of Environmental Sciences, University of East Anglia , Norwich , UK
6 Departamento de Sistemática e Ecologia, Universidade Federal da Paraíba , João Pessoa , Brazil
Nelson David
Electronic publication date: 2020 Aug 27
Publication date: 2020
Volume: 8
Electronic Location ID: e9756
Received 2020 Mar 25; Accepted 2020 Jul 28
Copyright: © 2020 Miranda et al.
Copyright year: 2020
Copyright holder: Miranda et al.
License: This is an open access article distributed under the terms of the Creative Commons Attribution License, which permits unrestricted use, distribution, reproduction and adaptation in any medium and for any purpose provided that it is properly attributed. For attribution, the original author(s), title, publication source (PeerJ) and either DOI or URL of the article must be cited.
License URL: https://creativecommons.org/licenses/by/4.0/

Keywords: Bradypus, Canopy, Choloepus, Foraging, Seasonality, Apex predator, Tropical forest, Deciduousness, Moonlight, Harpia harpyja

Funding: United States Agency for International Development (USAID) Disney Worldwide Conservation Fund Wolf Creek Charitable Foundation Liz Claiborne and Art Ortenberg Foundation National Environmental Authority of Panama (ANAM, at present MiAmbiente) Rufford Small Grants Foundation 18743-1 and 23022-2 Cleveland Metroparks Zoo and SouthWild.com Peugeot-ONF Carbon Sink Reforestation Project University of KwaZulu-Natal (ZA) National Research Foundation (ZA) This work was supported by the donors and staff who have participated in the Peregrine Fund’s Harpy Eagle Restoration Program, including: United States Agency for International Development (USAID), Disney Worldwide Conservation Fund, Wolf Creek Charitable Foundation, Liz Claiborne and Art Ortenberg Foundation and National Environmental Authority of Panama (ANAM, at present MiAmbiente). Everton Miranda thanks the financial support of Rufford Small Grants Foundation (18743-1 and 23022-2), Rainforest Biodiversity Group, Idea Wild, The Explorers Club Exploration Fund—‘Mamont Scholars Program,’ Cleveland Metroparks Zoo and SouthWild.com. Everton Miranda logistical support was given by the Peugeot-ONF Carbon Sink Reforestation Project. Colleen T. Downs received funding from the University of KwaZulu-Natal (ZA) and the National Research Foundation (ZA). There was no additional external funding received for this study. The funders had no role in study design, data collection and analysis, decision to publish, or preparation of the manuscript.

==============================
Background

Climate plays a key role in the life histories of tropical vertebrates. However, tropical forests are only weakly seasonal compared with temperate and boreal regions. For species with limited ability to control core body temperature, even mild climatic variation can determine major behavioural outcomes, such as foraging and predator avoidance. In tropical forests, sloths are the arboreal vertebrate attaining the greatest biomass density, but their capacity to regulate body temperature is limited, relying on behavioural adaptations to thermoregulate. Sloths are largely or strictly nocturnal, and depend on crypsis to avoid predation. The harpy eagle (Harpia harpyja) is a sloth-specialist and exerts strong top-down control over its prey species. Yet the role of environmental variables on the regulation of predator–prey interactions between sloths and harpy eagles are unknown. The harpy eagle is considered Near Threatened. This motivated a comprehensive eﬀort to reintroduce this species into parts of Mesoamerica. This eﬀort incidentally enabled us to understand the prey profile of harpy eagles over multiple seasons.

Methods

Our study was conducted between 2003 and 2009 at Soberanía National Park, Panamá. Telemetered harpy eagles were seen hunting and feeding on individual prey species. For each predation event, field assistants systematically recorded the species killed. We analysed the effects of climatic conditions and vegetation phenology on the prey species profile of harpy eagles using generalised linear mixed models.

Results

Here we show that sloth predation by harpy eagles was negatively aﬀected by nocturnal ambient light (i.e. bright moonshine) and positively aﬀected by seasonally cool temperatures. We suggest that the first ensured low detectability conditions for sloths foraging at night and the second posed a thermally unsuitable climate that forced sloths to forage under riskier daylight. We showed that even moderate seasonal variation in temperature can influence the relationship between a keystone tropical forest predator and a dominant prey item. Therefore, predator–prey ecology in the tropics can be modulated by subtle changes in environmental conditions. The seasonal eﬀects shown here suggest important demographic consequences for sloths, which are under top-down regulation from harpy eagle predation, perhaps limiting their geographic distribution at higher latitudes.

Introduction

Predation is a central theme in ecology and evolution, driving morphological, physiological, and behavioural responses in prey species to the threat of death or injury (Genovart et al., 2010). Both the nature and magnitude of predation as a dominant ecological force are affected by seasonality (Darimont & Reimchen, 2002). However, the seasonality of predator–prey relationships in tropical forests is at best considered to be subtle compared with temperate and boreal regions, because of the comparatively low variation in day length and ambient temperature (Forsythe et al., 1995). Nevertheless, tropical forests can experience considerable seasonality in leaf flushing and fruiting as a response to climatic variables (Mendoza, Peres & Morellato, 2016). While available data suggests that climatic conditions in tropical environments have strong effects on animal activity (Foster et al., 2013; Cid, Oliveira-Santos & Mourão, 2015), there are relatively few studies about the nature of such effects on predator–prey interactions.

Seasonally elevated rainfall and the resulting responses in vegetation growth can provide food and cover for many arboreal taxa in tropical forests (Haugaasen & Peres, 2009). Conversely, the dry season often induces leaf abscission in trees and woody lianas (Souza, Gandolfi & Rodrigues, 2014), which may limit food availability and shelter to arboreal folivores. The combination of reduced cover and limited food resource availability can enhance predation risk (Menezes, Kotler & Mourão, 2014; Menezes, Mourão & Kotler, 2017). The seasonal variation may modify the range of thermal microhabitats available to a prey species. As endothermic forest specialists, sloths (genus Bradypus and Choloepus, order Pilosa) exhibit relatively low basal metabolic rates and can only partially regulate body temperature (Pauli et al., 2016). Therefore, they need to bask and can be affected by even mild variation in habitat cover and thermally inappropriate microhabitats (Peery & Pauli, 2014; Giné et al., 2015), to the extent that temperature seasonality is highly influential on sloth behavioural ecology (Moreira et al., 2014).

Sloths from the Bradypus and Choloepus genus differ in their biology. Choloepus are more vigorous (Pauli et al., 2016), larger (~6 kg; Wetzel & Montgomery, 1985), have a higher body temperature (Vendl et al., 2016), and a more diversified diet (Dill-McFarland et al., 2016). Bradypus sloths fit the stereotypical sluggish behaviour of sloths (Pauli et al., 2016), are smaller (~4 kg; Wetzel & Montgomery, 1985), have a relatively low body temperature (Vendl et al., 2016), and feed on leaves exclusively (Dill-McFarland et al., 2016). Finally, two-toed sloths (Choloepus spp.) are nocturnal, whereas three-toed sloths (Bradypus spp.) are cathemeral (Sunquist & Montgomery, 1973; Giné et al., 2015).

Likewise, moonlight is likely to alter animal behaviour by affecting detectability of both predators and prey at night (San-Jose et al., 2019). Lunar phobia by mammals is widely justified as a strategy to prevent predation (Cozzi et al., 2012). However, a metanalysis by Prugh & Golden (2014) showed that the response to lunar light was typically idiosyncratic. While visually-oriented mammals have an increased activity response to lunar light, mammals that have weak vision—like sloths—generally decrease activity on bright nights (Prugh & Golden, 2014) and therefore are less likely to suffer predation.

We can expected that the seasonality of predator–prey relationships involving sloths might be affected by even subtle climatic fluctuations in ambient temperature. Sloths are important prey species that rely heavily on crypsis to avoid predation, rather than evasive responses once they are detected (Touchton, Hsu & Palleroni, 2002). However, studies attempting to identify the cues leading to seasonal changes in prey activity and predation are inherently hindered by small sample sizes. While apex predators have profound effects on ecosystem structure and function (Terborgh et al., 2001), they are difficult to study, rendering this lack of knowledge almost impossible to overcome.

The harpy eagle (Harpia harpyja; Fig. 1) is considered Near Threatened by the IUCN (Birdlife International, 2017), mainly because of human persecution (Muñiz-López, 2017) and habitat loss, which have extirpated these mega-raptors from 41% of their former historical range distribution (Miranda et al., 2019). Harpy eagles are an apex predator that specialises on sloths, relying heavily on these prey species wherever they co-occur (Aguiar-Silva, Sanaiotti & Luz, 2014; Miranda, 2015). Harpy eagles hunt passively by visually scanning and listening to the forest canopy (Touchton, Hsu & Palleroni, 2002). They are unique among eagles having a large retractable facial disc to enhance their hearing (Ferguson-Lees & Christie, 2001). Harpy eagles are the Earth’s largest eagles. Being large-sized, they can prey on sloths of any age (Aguiar-Silva, Sanaiotti & Luz, 2014), including adult individuals of all continental sloth species (Miranda, 2018). Harpy eagle-sloth predator–prey systems are therefore ideal candidates to investigate how changes in climate and moonlight may affect multispecies predation rates. The Peregrine Fund has lead a comprehensive effort to reintroduce this species into parts of Mesoamerica (Campbell-Thompson et al., 2012; Watson et al., 2016). This effort, spanning from 2003 to 2009, incidentally enabled us to understand, for the first time, the prey profile of harpy eagles over multiple seasons.

Figure 1 Harpy eagle preying over sloth.

Adult female harpy eagle (Harpia harpyja) eating a young Two-toed sloth (Choloepus didactylus; Photo: Danilo Mota).

We explored environmental determinants of prey capture rates of reintroduced harpy eagles in Soberanía National Park (SNP); a tropical protected area in Panamá. Our goals were twofold: (1) to assess the effects of seasonality—like temperature, rainfall and leaf decidousness—on sloth capture rates by harpy eagles; and (2) to assess how moonlight could affect sloth and nocturnal prey predation rates. We predicted that: (1) sloth predation rates would increase with low temperatures, high rainfall and low leaf cover; (2) sloth and nocturnal prey predation rates would increase with low moon brightness.

Materials and Methods

Study site

Our study was conducted between 2003 and 2009 at Soberanía National Park (hereafter, SNP), a 19,545 ha protected area in eastern Panama along the banks of the Panama Canal (9°07′13″N, 79°39′37″W). The vegetation of SNP consists of semi-deciduous, seasonally moist tropical forest, most of which is now advanced (>80 years) secondary forest (Bohlman, 2010). The area has most of the staple prey species targeted by harpy eagles (Aguiar-Silva, Sanaiotti & Luz, 2014), including three-toed sloths (Bradypus variegatus), Hoffman’s two-toed sloths (Choloepus hoffmanni), white-nosed coati (Nasua narica), northern lesser anteater (Tamandua mexicana) and mantled howler monkeys (Alouatta palliata), all of which are either strictly arboreal or scansorial mammals. The Peregrine Fund had conducted experimental harpy eagle releases within SNP since 1997 (Muela et al., 2003; Watson et al., 2016), therefore we assumed that none of the prey species here were predator-naïve during our study.

The SNP has a marked dry season from December to April and a wet season from May to November. The wet season concentrates 85.3% of the annual rainfall, which averaged 2,242 mm p.a. for 2003–2009. During the dry season, the mean, minimum and maximum ambient temperatures were 27.3 °C, 22.1 °C, 33.0 °C, respectively, and slightly warmer than the corresponding temperatures during the wet season (26.5 °C, 23.2 °C, 30.9°C, respectively). Daily climate data were obtained from ETESA (http://www.hidromet.com.pa/), using Hodges Hill Meteorological Station data for rainfall (15 km from the release site) and the Tocumen Station for data on temperature (43 km from the release site). A Walter-Lieth climate diagram describing the seasonality of rainfall and ambient temperature in the park can be seen in Fig. S1.

Harpy eagle prey profile

Before final release, captive-bred harpy eagles were soft-released at SNP by a process known as hacking (Muela et al., 2003). This allowed harpy eagles to learn how to hunt, as would occur in the wild (Muñiz-López et al., 2016). Further details on the harpy eagle reintroduction protocols and results are available in Campbell-Thompson et al. (2012) and Watson et al. (2016). Harpy eagles were fitted with both radio-telemetry and GPS tags. During soft releases, they were fed thawed rats and rabbits, always using a blind to avoid food conditioning with humans. Foraging independence was defined on the basis on an eagle being able to make two unassisted successive kills within 20 days or survive 30 days without food provisioning, thereby demonstrating that it was able to hunt self-sufficiently. Both regular radio- and global position system (GPS)-tracking leading to visual contact with each telemetered eagle was required to check its condition.

As the reintroduced hapy eagles were captive-born sub-adults (5–22 months; Campbell-Thompson et al., 2012) from captive stock maintained by The Peregrine Fund, we performed an a priori graphical analysis to ensure that the diet of reintroduced harpy eagles was similar to that of wild adult individuals. We did so by dividing the number of captured prey items within blocks of 25 samples (which adequately represents the main prey species; Miranda, 2015) and distributed them according to ontogeny or experience. We defined ontogeny as age in months for any given predation event, whereas we defined experience as any given predation event relative to the number of days since the first wild prey item was captured. Neither ontogeny nor experience affected harpy eagles’ patterns of predation as there was no evidence of nested patterns that would be expected if shifts in prey preferences occurred (Figs. S2 and S3). We therefore consider hunting patterns by reintroduced harpy eagles comparable with those of wild adults, and this was consistent with previous reports (Touchton, Hsu & Palleroni, 2002). The spatial distribution of those kill sites, as well as the location of the release site and meteorological stations within SNP are shown in Fig. 2.

Figure 2 Study site.

Location of Soberanía National Park in central Panama (lower left inset map), showing the location of 189 predation events (green dots), release site (white star) and meteorological stations (white triangles).

Predation and environmental determinants

During observations, while tracking, harpy eagles were seen hunting and feeding on individual prey species. For each predation event, field assistants systematically recorded all species killed (whenever identification to the level of species was possible). Field assistants were instructed to remain as inconspicuous as possible and leave the eagles alone as soon as observations were recorded. Prey items of known species identity were recorded during all months of the year, over the 7-year study, although observations were typically sparser during the month of November.

We related measures of climatic seasonality and vegetation phenology to the prey species profile of harpy eagles. Daily climatic data on precipitation and ambient temperature, were obtained from nearby meteorological stations. Data on the phases of the lunar cycle at a daily resolution over the entire study period were obtained from http://www.astronomyknowhow.com. We used the percentage of moon shade cover per night as a proxy for light availability. We used the normalised difference vegetation index (NDVI) as a proxy for canopy leaf deciduousness, where NDVI = (IR − R)/(R + IR), IR being the near-infrared LandSat band 4 and R the red LandSat band 3. NDVI values were calculated using georeferenced LandSat images obtained for all months of the year during the study period. NDVI is a measure of vegetation ‘greenness,’ rather than deciduousness, but is highly correlated to leafing cycles (Bohlman, 2010). For each prey detection event, we estimated the NDVI score of all 30 m × 30 m pixels within a 1 km radius of the location of each predation event for the nearest five dates of LandSat images available for that period. We then interpolated these indices to estimate the composite NDVI metric for the detection date of each prey item.

We ran two batches of generalised linear mixed-effects models (GLMM) using as response variables (1) the probability of any given prey item being a sloth (either Bradypus or Choloepus) and (2) the probability of any given prey item being nocturnal. Because the set of environmental covariates for each model was large, we used a backwards AIC-based stepwise algorithm to select the most important variables for each fixed-effect model, adding the random effect afterwards. All GLMMs were run using a binomial error structure and the logit link function, and bird identity as a random effect on the intercept. All variables used were checked for covariance using the Variance Inflation Factor (VIF). All analyses were run using the R 3.6.1 platform. Environmental covariates used in each GLMM are presented in Table S1. All source codes used in the analyses are available at https://github.com/KenupCF/HarpySlothPredation.

The Peregrine Fund Harpy Eagle Restoration Program complied with the laws of Panamá during the time in which the project was performed, with permits granted by National Environmental Authority of Panama (ANAM, at present MiAmbiente and SISBIO #58533-5).

Results

We recorded a total of 200 harpy eagle predation events, from which we obtained positional data for 189 prey items, 173 of which were identified. These prey items were killed by 33 harpy eagles during six dry seasons and six wet seasons during the 7 years of study. This amounted to 88 prey samples during the dry seasons and 85 samples during the wet seasons. The temporal distribution of predation records and the functional groups of prey species showed that sloths were by far the most important prey species for harpy eagles (Fig. 3). Two sloth species represented 65.3% of the harpy eagle diet in terms of the overall numeric prey profile, of which brown-throated sloths, Hoffman’s two-toed sloths and unknown sloths represented 34.1%, 15.6% and 15.6% of all prey items, respectively. Second to sloths, the next most significant dietary contributors to harpy eagles were white-nosed coatis (7.5%), northern lesser anteaters (6.9%) and mantled howler monkeys (5.2%). Further information on the prey species composition is shown in Table 1.

Figure 3 Prey composition and effort.

Monthly distribution of harpy eagle kills throughout the year. Vertical bars are color-coded according to the main prey functional groups. Observations were made in all months of the year, however more scantly in November.

Table 1 Prey composition in the diet of harpy eagles.

Seasonal changes in incidence of kills by harpy eagles shown in percentages, combining frequencies for both wet and dry seasons across the seven years of study (2003–2009). Overall column shows percentages of prey items for all periods combined, and sample sizes (in parentheses). See “Study Site” section of Methods for further details of season definition.

Species	Dry %	Wet %	Overall % (n)	
Brown-throated sloth Bradypus variegatus	36.8	31.4	34.1 (59)	
Hoffmann’s two-toed sloth Choloepus hoffmanni	24.1	7.0	15.6 (27)	
Unidentified sloths	11.5	19.8	15.6 (27)	
White-nosed coati Nasua narica	5.7	9.3	7.5 (13)	
Northern lesser anteater Tamandua mexicana	2.3	11.6	6.9 (12)	
Mantled howler monkey Alouatta palliata	3.4	7.0	5.2 (9)	
Green Iguana Iguana iguana	4.6	2.3	3.4 (6)	
Common opossum Didelphis marsupialis	2.3	2.3	2.3 (4)	
White-headed capuchin Cebus capucinus	2.3	2.3	2.3 (4)	
Collared peccary Tayassu tajacu	1.1	2.3	1.7 (3)	
Nine-banded armadillo Dasypus novemcinctus	1.1	1.2	1.1 (2)	
Central American agouti Dasyprocta punctata	2.3	0.0	1.1 (2)	
Crab-eating raccoon Procyon cancrivorus	1.1	0.0	0.5 (1)	
Tayra Eira Barbara	1.1	0.0	0.5 (1)	
Black vulture Coragyps atratus	0.0	1.2	0.5 (1)	
Unidentified parrot	0.0	1.2	0.5 (1)	
Unidentified monkey	0.0	1.2	0.5 (1)	

Sloth predation rates increased significantly during low moon brightness (β = −0.648, p = 0.0116) and low ambient temperatures with marginal statistical significance (β = −0.508, p = 0.0535; Fig. 4). Harpy predation on nocturnal animals was weakly affected by low moon brightness (Fig. 4), but this lacked sufficient statistical significance (β = −0.392, p = 0.1461). Rainfall and leaf deciduousness had no discernible effect in any of our models. Statistical results are summarised in Table 2.

Figure 4 Effect of environmental variables on the probability of predation events by harpy eagles.

(A) Effect of moon brightness on sloth predation probability: fewer sloths were taken during bright moonlit nights (p = 0.0134). (B) Effect of minimum temperature on sloth predation probability: fewer sloths were taken under cooler conditions (p = 0.0413). (C) Effect of moon brightness on nocturnal mammal predation: fewer nocturnal prey were killed. During bright nights, but this lacked statistical significance (p = 0.12).

Table 2 Results of generalized linear mixed models of harpy eagle prey profile.

First model predicts probability that a given animal preyed by a harpy eagle is a sloth, while the second model predicts probability of prey being a nocturnal animal. Both models use a logit link due to the binomial natural of the data. Both models use tracked individuals and years sample as random effects over the intercept.

Model	Variable	Estimate	Standard error	p-Value	Random individual variance	Random yearly variance	
Sloth	Intercept	0.588	0.470	0.2109	1.001	0.513	
Lunar disc (%)	−0.648	0.257	0.0116	–	–	
Minimum temperature (°C)	−0.508	0.263	0.0535	–	–	
Night	Intercept	−0.933	0.422	0.0271	0.336	0.367	
Minimum lunar disc (3-day; %)	−0.392	0.269	0.1461	–	–	

Discussion

Although environmental conditions either increase prey vulnerability or provide an advantage to sit-and-wait and pursuit predators (Doody, Sims & Letnic, 2007; Prugh & Golden, 2014), little has been documented on this topic in closed-canopy tropical forest ecosystems. In harpy eagle-sloth predator–prey systems, we showed increases in sloth nocturnal activity under elevated moon brightness and cryptic behaviour during the day provided mechanisms of escaping detection by harpy eagles. We also showed an increase in predation rates under cool temperatures, which may induce further diurnal activity of sloths. Finally, we examined the roles of leaf flush and rainfall on harpy eagle prey choice, but neither had a detectable effect on sloth predation rates. These results pose interesting questions about the consequences of temperature and moon brightness to this keystone Neotropical forest predator and its dominant prey species.

Moonlight has been shown to have contradictory effects on nocturnal mammal activity patterns in terms of their antipredator strategies. Prey species that can detect predators visually and anticipate their attacks with evasive maneuvers may increase foraging activity under high levels of moonlight, whereas those that cannot decrease activity (Prugh & Golden, 2014). Sloths, however, typically prefer to sleep at night in environments where they evolved with predator presence (Voirin et al., 2014), and in other areas generally showing greater fear of diurnal predators as harpy eagles. Indeed, there is anecdotal evidence of increased sloth activity during full moon phases (Beebe, 1926). Sloths are known to be lethargic and have extremely poor vision, while harpy eagles typically attack from distances of less than 30 m during daylight (Touchton, Hsu & Palleroni, 2002). We, therefore, expected that sloths reduce their overall activity during the day, instead foraging at night under bright moonlit to reduce predation risk, which significantly reduces the probability of successful attacks by diurnal harpy eagles. Success rates of harpy eagles predation on sloths is generally high compared with visually oriented prey: 55% of all attacked sloths are successfully killed, while only 33% of visually oriented prey are successfully killed if they had been attacked (Touchton, Hsu & Palleroni, 2002). This may be the underlying adaptive reason why sloths are inactive during the day if bright nights are available as foraging time, neutralising search images of diurnal predators and greatly reducing their detection probability by harpy eagles. Further sloth telemetry studies would provide confirmatory evidence.

In addition to the reduced predation levels of sloths during bright moon nights, we showed that as ambient temperatures increased, predation rates declined. Presumably, this happened because of the increased daytime activity levels of this endotherm, which is prone to metabolic torpor under cooler weather conditions, especially at night (Giné et al., 2015). It has been shown, for instance, that the nocturnal activity of the maned sloth (Bradypus torquatus) is inhibited by lower ambient temperatures (Chiarello, 1998). Predation rates of sloths by harpy eagles were higher during colder conditions, which likely induce compensatory activity by sloths during the warmer daytime. Basking behavior of sloths increases with lower ambient temperatures along altitudinal gradients in mountainous areas (Urbani & Bosque, 2007). Another possible explanation for the temporal changes in sloth predation rate could result from its reproductive behaviour. However, the literature shows weak and idiosyncratic evidence for seasonal breeding for both sloth species present in our study area (Taube et al., 2001). These features reinforce our premise that behavioural crypsis is the main antipredator strategy of sloths, which we suggest to be the underlying reasons for the patterns observed in our study. Indeed, the latitudinal boundaries of the geographic distribution of sloths are far more restricted than those of harpy eagles (Moreira et al., 2014; Miranda et al., 2019). Sloths of the Choloepus genus are distributed over tropical Central America and the pan-Amazonian region, while Bradypus also occur over the northern section of Atlantic Forest (Emmons & Feer, 1997). Predation by harpy eagles may play a key role in limiting sloth geographic distribution—and altitudinal ranges—given that sloths would be required to compensate for cooler temperatures in the southern Atlantic Forest or higher regions by increasing levels of diurnal activity (Chiarello, 1998; Urbani & Bosque, 2007). Therefore, this would inhibit extended periods of inactivity induced by cool temperatures, but increase temporal activity overlap with diurnal predators.

Rainfall apparently had no effect in any of our models explaining the incidence of sloth predation, a pattern that could also be explained by low predation risk resulting from the cessation of harpy eagle activity during rainy weather (Touchton, Hsu & Palleroni, 2002), or even distance from the meteorological stations, inducing error. Leaf abscission presented no effects on predation of sloths. Although we predicted increased probability of arboreal prey detection under leafless conditions in the semi-deciduous forests of central Panama, forest areas dominated by leafless trees and/or woody lianas may be consistently avoided by prey species relying on concealed foraging activity (Menezes, Kotler & Mourão, 2014; Menezes, Mourão & Kotler, 2017). For a sloth, leafless tree crowns offer little if any protective cover and no food resources. Our robust methods to estimate levels of deciduousness combined with a wide buffer describing the likely sight range of potential kills suggest that arboreal habitats lacking foliage cover would be avoided not only by prey species but also by harpy eagles, thereby at least partly explaining why deciduousness had no effects in any of our models.

Nocturnal prey capture by harpy eagles was not significantly affected by any of the environmental covariates, and the fact that these large diurnal raptors can frequently successfully kill several strictly nocturnal prey species remains puzzling. Modest increases in predation rates of nocturnal mammals were associated with darker nights, when nocturnal species typically preyed by harpy eagles (anteaters, opossums and armadillos) are expected to be more active given their poor ability to anticipate incoming predators visually (Caro, 2005; Prugh & Golden, 2014). The harpy eagle sit-and-wait predation strategy is further enhanced by their retractable facial disc, which performs the same function as in strictly nocturnal raptors (i.e. owls), of improving acoustic detection of prey. Combined with extremely acute vision, which is likely associated with a high density of photoreceptor cells in the retina typical of many diurnal raptors (Lisney et al., 2013), harpy eagles are superbly capable of locating inconspicuous prey, enabling them to be the only Neotropical apex predator to specialise on the highly secretive sloths (Miranda, 2015; Miranda, Menezes & Rheingantz, 2016). Harpy eagle activity patterns can be investigated with further research using either intensive telemetry-assisted follows or camera trapped nests. By including nocturnal telemetry or motion-sensitive telemetry devices on monitoring schedules or confirming that harpy eagles can deploy crepuscular/nocturnal hunting effort at the time of nesting (e.g. evidenced by nocturnal prey delivery) would largely solve this question.

Our results suggest important consequences for patterns of prey mortality through the tropical seasons of Neotropical forests. We, therefore, suggest that researchers, conservationists and practitioners can learn from natural fluctuations in predator–prey systems when designing management actions (such as reintroduction, release and translocation efforts) of both harpy eagles and their prey, since some of these prey species are also threatened (Catzeflis et al., 2008; Moreira et al., 2014; Suscke et al., 2016). For instance, consequences of the harpy eagle reintroduction on the endemic maned sloth which is listed as Vulnerable in the Brazilian Atlantic Forest needs careful evaluation.

Conclusions

We showed that the probability of harpy eagles preying on sloths decreased in response to nocturnal high moon brightness and increased with low temperatures. This almost certainly occurs because sloths respond to low temperatures foraging more in the daytime, and circumvent high diurnal detectability by foraging on bright moonlit nights when they are not exposed to visually oriented predators. These conceptually simple conclusions result from overcoming the formidable challenges of monitoring the diet of apex predators in tropical forests for extended periods. We further note that the seasonal effects we uncovered here suggest important consequences for herbivore prey species, whose populations are likely regulated by top-down predation from harpy eagles and other top predators. The magnitude of cyclic changes in predator–prey interactions shown here potentially are even stronger in more seasonal tropical and subtropical forests experiencing cooler seasons, higher altitudes or prolonged flood pulses. Further studies on a diverse set of predator and prey assemblages in tropical forests elsewhere would help fill this knowledge gap.

Supplemental Information

Supplemental Information 1 Raw data.

Date: day, month, year; Prey ID: species; Day: day; Month: month; Year: year; Age_class: prey age estimate, one is adult and two is juvenile; ID: predation event individual number; UTM_E: geographic coordinates; UTM_N: geographic coordinates; NDVI_1: deciduousness; NDVI_5: deciduousness; Rainfall: rainfall; Max_temp: maximum temperature; Min_temp: minimum temperature;

moon.yest: moon in the day before; moon.max: maximum moon brightness; rfall.sum: precipitation sum for three days;

max_temp.max: maximum temperature in three days; min_temp.min: minimum temperatures in three days; Sloth: if the prey is a sloth; Night_prey: if the prey is nocturnal; is.brad: if it is from Bradypus genus is.chol: if it is from Choloepus genus

Click here for additional data file.

Supplemental Information 2 Climatic variables used in mixed generalized linear models (GLMM) explaining the dietary profile of harpy eagles.

Each GLMM is used to predict the probability that a given prey item belongs to the response variable. ‘Normalized difference vegetation index’ (NDVI) is a proxy for deciduousness, calculated using data from LANDSAT imagery. All other climatic variables were obtained from meteorological stations near the study site. Variables noted as ‘three-day’ are pooled over a three-day period up to any given prey detection event.

Click here for additional data file.

Supplemental Information 3 Climate diagram.

Walter-Lieth diagram showing precipitation and temperatures at the study site.

Click here for additional data file.

Supplemental Information 4 Correlogram showing the lack of effects of experience in prey catching patterns.

Number and species of individuals captured along ageing in reintroduced harpy eagles.

Click here for additional data file.

Supplemental Information 5 Lack of effect of experience in reintroduced harpy eagle prey composition.

Harpy eagle experience in days and prey number and species. No effect of greater experience can be seen in prey composition.

Click here for additional data file.

We thank the many institutions and staff who have participated in The Peregrine Fund’s Harpy Eagle Restoration Program. We also thank hack-site attendants for field assistance. Everton Miranda’s logistical support was given by the Peugeot-ONF Carbon Sink Reforestation Project, based at the São Nicolau Farm in Cotriguaçu, Mato Grosso, Brazil. This Project is a Peugeot initiative to fulfil some of the Kyoto Protocol directions and is run by the ONF-Brasil enterprise. Marcus Canuto, Jose Vargas and Alexander Blanco provided useful discussions, improving a previous version of the manuscript. The Peregrine Fund Harpy Eagle Restoration Program complied with the laws of Panamá during the time in which the project was performed. We are grateful for the constructive comments of Mario Fernando Garces Restrepo and an anonymous reviewer.

Additional Information and Declarations

Competing Interests

Author Contributions

Animal Ethics

Data Availability

Félix Hernán Vargas and Richard Watson are employed by The Peregrine Fund. Edwin Campbell-Thompson and Angel Muela were employed by The Peregrine Fund.

Everton B.P. de Miranda conceived and designed the experiments, analyzed the data, prepared figures and/or tables, authored or reviewed drafts of the paper, and approved the final draft.

Caio F. Kenup conceived and designed the experiments, analyzed the data, prepared figures and/or tables, authored or reviewed drafts of the paper, and approved the final draft.

Edwin Campbell-Thompson performed the experiments, authored or reviewed drafts of the paper, and approved the final draft.

Felix H. Vargas performed the experiments, authored or reviewed drafts of the paper, project administration, fund raising, and approved the final draft.

Angel Muela performed the experiments, authored or reviewed drafts of the paper, and approved the final draft.

Richard Watson conceived and designed the experiments, authored or reviewed drafts of the paper, project administration, fund raising, and approved the final draft.

Carlos A. Peres performed the experiments, prepared figures and/or tables, authored or reviewed drafts of the paper, and approved the final draft.

Colleen T. Downs conceived and designed the experiments, prepared figures and/or tables, authored or reviewed drafts of the paper, language review, and approved the final draft.

The following information was supplied relating to ethical approvals (i.e. approving body and any reference numbers):

The Peregrine Fund Harpy Eagle Restoration Program complied with the laws of Panamá during the time in which the project was performed, with permits granted by National Environmental Authority of Panama (ANAM, at present MiAmbiente) (SISBIO #58533-5).

The following information was supplied regarding data availability:

Data is available as a Supplemental File. Codes are available at GitHub: https://github.com/KenupCF/HarpySlothPredation.

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
