# Peer review of "High moon brightness and low ambient temperatures affect sloth predation by harpy eagles"

_PeerJ, doi:10.7717/peerj.9756_

## Round 0.1 · original submission · Major Revisions

Both reviewers appreciated the unique nature of this study and dataset. However, both reviewers also had concerns about the clarity and organization of the manuscript, as well as the need for improvement in how the data were analyzed and interpreted. Please address each of the comments from the reviewers in your revisions.

Reviewer 1 ·

Basic reporting

Although the manuscript is well written and clear, there are some sections that should be revised. A major concern is that the introduction and discussion are not written in the same way. While the introduction focuses on the prey, the discussion focuses on the predator. I suggest authors review these sections. Furthermore, a variable that seems to be important in the predator-prey system, such as the brightness of the moon, is only mentioned very slightly in the introduction.
On the other hand, I suggest that the methods section in the abstract have to be revised, you should explain better the lines 60-61.
After reviewing the data and the script, I suggest the authors send Table 1 to the supplementary material and replace it with a table showing all the final models and their variables. Table 2 and Figure 3 show information regarding the same issue. I suggest authors try to synthesize the information into one.

Experimental design

Despite the fact that the research is original and delves into a very interesting and not studied topic such as harpy eagle-
sloth predator-prey systems I think the working hypotheses are somewhat confusing. In the introduction, different approaches to the proposed hypothesis are evaluated. The objectives of the study are not clear and the last paragraph is somewhat confusing. I suggest reviewing this section. I have some comments to help improve the manuscript:

1. The material and methods and results section don´t have the same framework. The authors explain several subsection within the material and methods section, which are not reflected in the results. Also the subsections within the methods are somewhat messy. I suggest checking the HARPY EAGLE PREY PROFILE subsection, as it mixes tagging GPS information and previous experiments. I suggest including more information about the type of gps device, method of placement, type of programming to get data .


Minor comments:
ln133: falconary is not a properly term here

ln 137: What is the age of the eagles? Born this year? please specify

ln142-146: I suggest include in the results section. I don't understand why it compares to a water bird. I suggest deleting the quote (Daunt F, WANLESS S, HARRIS MP, Money L, Monagham P, MONAGHAN P. 2007. Older and wiser: improvements in breeding success are linked to better foraging performance in European shags. Functional Ecology 21: 561–567 ) and look for a more suitable one, for example other raptors in reintroduction programs.

Validity of the findings

I thank you for providing the raw data and the script. After a review I suggest the authors include the year variable as a random factor in the models to check if there is an annual variation.
The authors' conclusions are consistent with the results, although as they indicate, it would take a few more studies to verify their final hypothesis.

Additional comments

The manuscript addresses some important ecological issue related with the harpy eagle-
sloth predator-prey system. While the study topic is interest and contribute to the knowledge about an endangered raptor I found the manuscript to be somewhat confusing and lacking some details.
I hope the suggested changes help improve this manuscript.

·

Basic reporting

The data, analysis, writing, review of literature and images are very good. But part of the analysis and some explanations of some biological definitions must be restructured.

Experimental design

It is a very good design of experiments, however it would be important to take the species Bradypus variegatus and Choloepus hofmanni separately because they show great differences in their behavior.

Validity of the findings

Supporting some conclusions in anecdotal facts can be very dangerous. It is well known that the species of the genus Choloepus are nocturnal, but assume that there is significant nocturnal activity of Bradypus due to an article by an individual defecating is not good, in my experience in radio-tracking in Colombia and Costa Rica nocturnal activity in Bradypus variegatus is almost non-existent.

"we show that as 240 temperatures increase, predation rates decline". An explanation for this pattern is that Bradypus variegatus will spend less time basking in the sun and therefore exposed when ambient temperatures are higher. In articles cited in the manuscript it is explained how the temperature of Bradypus vaariegatus varies as the environmental temperature does (Pauli et al 2016), for its activation and activation of their microbiota this species has a basking behavior, which can vary in duration depending on the body temperature reached by the species, in some mountainous localities where the environmental temperature is lower this behavior may have a longer duration (Urbani & Bosque 2007).

Finally, the explanation of lunar phobia have to improve, several articles exhibit that in dark nigths the activity of nocturnal mammals increase because the amount of enviromental ligth dcrease, thus, the predation risk.

Additional comments

The current manuscript has an incredible data set with which many analyzes of the predation by the arpya eagle and its relationship with its prey could be made; however, there are major problems in the analysis and structure of the document.

First, the differences in the natural history of the sloths species of the genus Bradypus and the denus Choloepus are unknown or despised, For example the differnces at the level of daily activity and metabolic needs.

On the other hand, there are some flaws in the structuring of the introduction where important topics for the development of the manscript are omitted such as lunar phobia and the general biology of the arpya.

Third, an analysis taking the sloths of the genus Bradypus separately could clarify some of the effects of the variables on the predation of sloths.

Finally, there is a management of the term specialization that may be confused, one of the evidences of specialized animals in a prey is that its geographical distribution is less than the distribution of the resources in which it specializes, contrary to what was reported by the authors in this manuscript.

---

## Round 0.2 · Minor Revisions

The manuscript has been re-reviewed by one of the original reviewers. This reviewer notes some minor issues, which I would ask the authors to consider. I also noted some minor editorial issues (listed below) from my own reading of the manuscript.

Lines 38-39: This text isn’t methods. Please move it elsewhere.

Lines 40-42: This text isn’t methods. Please move it elsewhere.

Line 190: I believe “snormalised” should be “normalized”.

Line 199: I believe “sgeneralised” should be “generalised”.

Line 247: “manoeuvres” should be “maneuvers”.

Line 250: insert a space after “generally”.

Line 261: I believe “sneutralising” should be “neutralising”.

Line 291: I believe “spredicted” should be “predicted”

Reviewer 1 ·

Basic reporting

The new version of the manuscript is much clearer and better structured. The authors have included the changes suggested by both reviewers and this can be seen in the present manuscript. However, I think the introduction could still be improved. I suggest that the authors review the paragraph that talks about the Harpy Eagle (108-123). They should talk about the eagle and its status first and then the predator prey- system.

In the previous review, I suggest the authors send Table 1 to the supplementary material and replace it with a table showing all the final models and their variables. They response that accept this suggestion; the new table has been added as suggested.However, the same table appears in the pdf file as in supplementary material and not the one I suggested. The autor should review thats Table.

However there is a misstmach between text and tables. In the text, the authors say “ Statistical results are summarised in Table 2”, but Table 2 is related with “Prey composition in the diet of harpy eagles”. They should review that issue.

Experimental design

Now the objectives and the Hypotheses are clear. The authors improve the methods section. They included all the suggestion of the previous review.

Validity of the findings

No comment

Additional comments

The manuscript has improved since the previous revision. However, the authors can improve the manuscript. I have suggested some small changes that I hope the authors appreciate.

---

## Round 0.3 · accepted · Accept

Thanks for your thorough responses to the reviewer feedback.